# Role of Transcriptional and Epigenetic Regulation in Lymphatic Endothelial Cell Development

**DOI:** 10.3390/cells11101692

**Published:** 2022-05-19

**Authors:** Hyeonwoo La, Hyunjin Yoo, Young Bin Park, Nguyen Xuan Thang, Chanhyeok Park, Seonho Yoo, Hyeonji Lee, Youngsok Choi, Hyuk Song, Jeong Tae Do, Kwonho Hong

**Affiliations:** Department of Stem Cell and Regenerative Biotechnology and Institute of Advanced Regenerative Science, Konkuk University, Seoul 05029, Korea; lahw94@gmail.com (H.L.); hyunjinyoo7@gmail.com (H.Y.); fyyhgg@hanmail.net (Y.B.P.); thang.nx1012@gmail.com (N.X.T.); chpark0729@gmail.com (C.P.); upreference98@naver.com (S.Y.); affogatojoa@gmail.com (H.L.); choiys3969@konkuk.ac.kr (Y.C.); songh@konkuk.ac.kr (H.S.); dojt@konkuk.ac.kr (J.T.D.)

**Keywords:** epigenetics, transcription factor, lymphatic endothelium, lymphatic disease

## Abstract

The lymphatic system is critical for maintaining the homeostasis of lipids and interstitial fluid and regulating the immune cell development and functions. Developmental anomaly-induced lymphatic dysfunction is associated with various pathological conditions, including lymphedema, inflammation, and cancer. Most lymphatic endothelial cells (LECs) are derived from a subset of endothelial cells in the cardinal vein. However, recent studies have reported that the developmental origin of LECs is heterogeneous. Multiple regulatory mechanisms, including those mediated by signaling pathways, transcription factors, and epigenetic pathways, are involved in lymphatic development and functions. Recent studies have demonstrated that the epigenetic regulation of transcription is critical for embryonic LEC development and functions. In addition to the chromatin structures, epigenetic modifications may modulate transcriptional signatures during the development or differentiation of LECs. Therefore, the understanding of the epigenetic mechanisms involved in the development and function of the lymphatic system can aid in the management of various congenital or acquired lymphatic disorders. Future studies must determine the role of other epigenetic factors and changes in mammalian lymphatic development and function. Here, the recent findings on key factors involved in the development of the lymphatic system and their epigenetic regulation, LEC origins from different organs, and lymphatic diseases are reviewed.

## 1. Introduction

The lymphatic system is involved in lipid reabsorption, fluid balance, and immune surveillance. In contrast to the blood circulation system, the lymphatic system, comprising the lymphatic vessels and lymph nodes (LNs), is blind-ended and unidirectional from the periphery to the heart. The vasculature of the blood circulatory system can only transport large molecules owing to the presence of the tightly connected zipper-like structures of the endothelial cell (EC) junctions [1]. The size limit of the molecules that can pass through the blood capillary is in the range of 5–12 nm; however, it can be 60 nm in some cases in the bone marrow [2]. Lymphatic EC (LEC) junctions in lymphatic capillaries exhibit button-like structures and conditional gating, which allows the transport of large molecules and cells with a size in the micrometer range [1,3,4,5]. This structural difference allows the lymphatic capillaries to complement the blood circulatory system by transporting and cycling back into circulation from the interstitial fluid the large and fat-soluble molecules which cannot be transported through the blood circulatory system. Furthermore, the physical separation of the blood and the lymph provides an environment that is minimally affected by the blood circulation and blood pressure exerted by the heart [6,7]. Lymphocytes and antigens in lymphatic organs must be attached to the epithelium of lymphatic organs to communicate through the signals involved in eliciting immune responses. The blood circulatory system transports oxygen and carbon dioxide. Thus, circulating blood creates a dynamic environment that is not optimal for the maturation of lymphocytes [8]. Hematopoietic stem cells reside in the bone marrow, thymus (where T-cell development occurs), LNs, tonsils, and spleen, providing an optimal environment for lymphocyte maturation and activation [7,9]. Therefore, the loss or impairment of lymphatic system function can lead to the development of various circulatory and immune-related diseases.

The major aims of treating various diseases are the restoration and enhancement of lymphatic function. Research on embryonic development is critical for devising therapeutic strategies for lymphatic diseases. The elucidation of the molecular mechanisms underlying the formation of the lymphatic system in the early developmental stages will enable the development of useful strategies for the reconstitution of the optimal functioning of the lymphatic system. Previous studies have utilized high-throughput sequencing (HTS) technology, genetic knockout (KO) and knock-in experiments, and lineage tracing to reveal the molecular interactions during lymphatic development. For example, genetic studies revealed that molecules such as VE-cadherin, a cell–cell adhesion molecule, and CCBE1, a factor involved in the activation of VEGFC, are essential for lymphangiogenesis [10,11,12]. Epigenetic factors are also critical mediators of the regulatory mechanisms of key transcription factors. Previous studies have reported the importance of epigenetic mechanisms in lymphatic development [5,13,14]. The mapping of chromatin dynamics using methods such as chromatin immunoprecipitation sequencing (ChIP-seq), DNase-seq, and assay for transposase-accessible chromatin-sequencing (ATAC-seq) has enabled detailed description of chromatin dynamics. Combination of transcriptome and epigenomic analyses, especially the analysis of chromatin conformation and histone modifications, could provide useful insights into the molecular mechanisms of lymphatic system development [13].

Additionally, recent studies have employed novel HTS and genetic alteration methods to identify key factors involved in LEC specification as well as to reveal the molecular characteristics of lymph capillaries, collecting vessels, valves, and various lymphatic organs, such as the LN, bone marrow, and spleen [14,15]. In contrast to the heterogeneity of the endothelium of the blood vessels, the organ-specific heterogeneity of LECs has only recently begun to be discussed. Additionally, the novel developmental origins of LECs have been recently discovered [3,14]. This review discusses transcriptional regulation, epigenetic regulation, and organ specificity during the development of the lymphatic system. Additionally, molecular alterations in clinical cases of lymphatic diseases have been outlined to provide insights into developing potential therapeutic strategies for lymphatic defects.

## 2. Diseases Associated with the Lymphatic System

Lymphatic vasculature dysfunction causes diverse pathological conditions, including lymphedema, inflammation, and neo-lymphangiogenesis, which can promote tumor metastasis [16]. Recent studies have reported the role of the lymphatic system in health and disease [3]. However, the genome-wide mechanisms of lymphatic disorders and the underlying pathogenesis have not been elucidated. This section discusses the current knowledge on lymphatic defects in humans and mice, along with their etiological and genetic factors.

### 2.1. Lymphedema

Congenital disorders in lymphatic network formation or lymphatic failure can potentially lead to lymphedema, owing to the stagnation of lymphatic circulation and the accumulation of fluid in the interstitial tissue [17]. Disfiguring and life-threatening diseases are characterized by leg swelling, tissue fibrosis, impaired immune responses, and fatty-acid degradation [18]. Lymphedema is categorized into primary (congenital) lymphedema and secondary lymphedema based on the etiology. The etiological factors for primary lymphedema are hereditary genetic mutations, while those for secondary lymphedema are infection, radiation damage, and postoperative complications. Various genetic mutations that induce primary lymphedema are summarized in Table 1 [3,19]. Mutations in key transcription factors, including *FOXC2*, *SOX18*, and *GATA2*, can cause primary lymphedema [20,21,22,23,24,25]. Most patients with primary lymphedema exhibit impaired lymphatic valve function and hypoplasia or hyperplasia of the lymphatic vessels. The most prevalent secondary lymphedema is caused by infection with parasites, such as *Wuchereria bancrofti* and *Brugia malayi* [26]. Infections from parasites that cause filariasis (known as elephantiasis) are common in tropical regions. Surgical removal of cancer tissue or radiation therapy can also induce secondary lymphedema by damaging lymphatic vessels and LNs. Approximately 20% of patients with breast cancer develop lymphedema because of the side effects of surgery or radiation [27]. Another potential molecular mechanism involved in the formation of lymphedema has been suggested by Díaz-Flores et al. Their recent studies have shown that during human LN development, intussusceptive lymphangiogenesis is induced by highly abundant and evenly distributed VEGFC [28]. In turn, intussusceptive lymphangiogenesis has been found to participate in the formation of the meshwork of processes in the LN sinuses. Their studies provided the foundation for the explanation of the role of intussusceptive lymphangiogenesis in clinical cases of lymphedema [28,29]. Furthermore, Ogino et al. showed that transplantation of adipose-derived stem cells accelerated LEC proliferation, increased lymphatic vessel numbers, and mitigated fibrosis of the surrounding interstitial tissue via intussusceptive lymphangiogenesis [30].

### 2.2. Lipid Homeostasis and Obesity

The lymphatic vasculature is involved in absorbing various nutrients and lipid molecules from the intestine. Lipid molecules are packaged into chylomicrons, which are absorbed in the gut villi and reabsorbed by mesenteric lymphatic vessels [50]. The lymphatic system is essential for regulating lipid metabolism and homeostasis. Dysfunctional mesenteric lymphatic vessels can lead to the accumulation of lipids in the abdominal cavity [51]. Mouse models of lymphatic disorders often exhibit accumulation of subcutaneous fat and abdominal chylous ascites or enhanced adipogenesis. Metabolic syndrome associated with obesity also leads to lymphatic anomalies [52,53,54]. The secretion of pro-inflammatory signals from the adipocytes can induce chronic inflammation and lymphatic dysfunction.

### 2.3. Inflammation

Lymphatic vessels enable the transportation of activated antigen-presenting cells to secondary lymph organ LNs during adaptive immune responses. In response to inflammatory stimuli, such as pro-inflammatory cytokines, activated immune cells exhibit upregulated expression of VEGFC [55,56] and enhanced lymphatic drainage [57]. The inhibition of VEGFR3 signaling results in lymphedema and prolonged immune responses after irradiation with UVB [58]. LECs also participate in the process of inflammatory response regulation by mediating antigen presentation and inducing CD4 T-cell tolerance [59,60]. Recent studies suggest that LECs present peptide:MHC-II complexes acquired from dendritic cells (DCs) [59,60,61] or participate in the process of antigen presentation of DCs by providing various peripheral tissue antigens (PTAs) to induce CD4 T-cell tolerance [61]. 

The role of lymphangiogenesis in transplant rejection is mediated through the CCL21/CCR7 pathway. In human kidney transplants, lymphatic vessels in the host tissue produce CCL21, attract CCR7-expressing DCs [62], and initiate adaptive immunity. However, the inhibition of VEGFR3 signaling downregulates CCL21 in transplanted LECs and impairs immune responses [63]. This suggests that lymphangiogenesis inhibitors are potential immunosuppressive agents.

### 2.4. Cancer

Skobe et al. demonstrated that lymphatic vessels serve as an avenue for metastasis during cancer progression [64]. Studies on animal models of cancer progression have reported that tumor cells secrete lymphangiogenic factors, including VEGFA, VEGFC, and VEGFD; induce active metastasis; and promote the circulation of cancer cells into LNs and other sites [65,66,67,68]. Treatment with VEGFA and VEGFC inhibitors decreased tumor metastasis into LNs and lungs in a mouse mammary-gland cancer model [69]. On the other hand, immune modulation by LV is also critical for the trafficking of DC and initiating anti-tumor adaptive immunity (i.e., T-cell responses), suggesting a dual function of lymphatics in tumor metastasis depending on tumor types and tumor progression [70,71].

In physiological conditions, the expression of VEGFR3 is restricted to LECs. However, VEGFR3 is expressed in malignant blood endothelial cells (BECs) during metastasis [72,73,74]. Moreover, VEGFR3 activity suppressed tumor angiogenesis in a mouse model. These studies indicate that the inhibition of lymphangiogenesis can potentially suppress tumor metastasis. On the other hand, recent studies have shown that VEGFC treatment with lymphatic expansion could enhance anti-tumor immunity and the efficacy of immunotherapy or radiotherapy for glioma, suggesting a dual function of the lymphatic system on tumor metastasis in a context-dependent manner [75,76,77].

### 2.5. Novel Functions of the Lymphatic System and Associated Diseases

Myocardial infarction (MI), a coronary artery disease, results from decreased blood flow caused by plaque formation in the interior arterial walls [78]. The accumulation of plaques containing lipids and cholesterols in the arteries leads to the narrowing of the arteries, which may result in heart injury, stroke, and atherosclerosis [79]. Recent studies have demonstrated that lymphangiogenesis induction can improve the prognosis of myocardial edema and inflammation caused by MI and delay atherosclerosis [80,81,82]. In the MI mouse model, VEGFC administration improved cardiac function by promoting lymphangiogenic responses [83]. Further validation was performed using a rat MI model. Sustained release of VEGFC_C152S_ resulted in the maintenance of fluid balance and the alleviation of fibrosis and cardiac inflammation in the rat MI model [80]. These studies suggest that the VEGFC–VEGFR3 pathway is a potential therapeutic target for cardiac diseases.

The brain tissue has an alternative fluid drainage system called the glymphatic system [84]. Meningeal lymphatic vessels (mLVs) are located in the dura mater on the surface of the brain along the dural sinus. Similar to the lymphatic system in other tissues, the glymphatic system promotes waste clearance and drains cerebrospinal and interstitial fluids of the brain into the mLVs [85,86]. A recent study demonstrated that mLVs, and not the dural sinuses, are the primary drainage system of cerebrospinal fluid [87]. Moreover, mLVs are reported to be involved in the pathogenesis of neurodegenerative diseases, such as Alzheimer’s disease (AD), Parkinson’s (PD) disease, and stroke. mLVs may play an important role in the clearance of toxic amyloid-beta and inflammatory mediators, which are reported to cause AD [84,88]. Treatment with VEGFC promotes brain lymphangiogenesis and glymphatic perfusion [89]. PD is characterized by impaired dopaminergic neurons and α-synuclein aggregation [90]. The blockage of mLVs in A53T mice overexpressing human α-synuclein resulted in a PD phenotype [91]. Furthermore, a stroke animal model exhibited a dysfunctional meningeal lymphatic system [92]. These studies suggest that targeting the meningeal lymphatic system may serve as an effective therapeutic strategy for neurodegenerative diseases.

In the eyes, Schlemm’s canals (SCs) are endothelium-lined channels that express both blood and lymphatic endothelial markers [93,94]. Proper functioning of the SC is required for draining the aqueous humor from the intraocular chamber and balancing ocular pressure [95]. Glaucoma, which is characterized by damage to the optic nerve, can lead to irreversible blindness. Increased intraocular pressure is one of the etiological factors for glaucoma. Decreased aqueous humor drainage in the SC increases ocular pressure and consequently leads to optic neuropathy [96]. The modulation of PROX1, VEGFR3, and TIE2 signals, which mediate SC development, can be a potential novel therapeutic strategy for glaucoma.

Lymphatic anomalies in patients with Crohn’s disease are characterized by a leaky lymphatic system at the inflamed intestinal wall and impaired drainage of LNs [97]. Tertiary lymphoid organs and B cell–rich structures were observed along with mesenteric collecting vessels in patients with Crohn’s disease. The involvement of lymphatic vasculature in pathological conditions has not been completely elucidated. Thus, further studies are needed to understand the mechanisms of lymphatic development and to devise novel therapeutic strategies.

## 3. Key Transcription Factors of LECs and Epigenetic Regulation of Their Transcription

### 3.1. Historical Aspects of Lymphatic Vessel Development

In 1902, Florence Sabin injected Indian ink into the jugular area of pig embryos and observed the primitive lymphatic organs connect with the blood vasculature in the cardinal vein [98]. The primitive lymphatic organ, called the ‘lymph sac,’ was assumed to be the origin of the lymphatic vessels. This was the first study to suggest that the lymphatic vasculature originated from the cardinal vein. Sabin’s findings have been validated using lineage-tracing experiments with LEC-specific markers and the cardinal vein origin of lymphatic vasculature is considered to be a widely accepted concept. Studies based on this concept have examined the development of the lymphatic system from the lymph sac to explain the mechanism of LEC differentiation. Srinivasan et al. used a lineage-tracing mouse model to demonstrate that LECs are differentiated from a subset of the endothelium in the cardinal vein. [99]. The markers for cells involved in LEC lineage specification include PROX1, LYVE1, NRP2, VEGFR3, and PDPN (Figure 1) [100,101]. Various studies have demonstrated that PROX1, is the ‘master regulator’ of LEC specification and maintenance [102]. PROX1 activity in the cardinal vein ECs upregulates the transcription of LEC-specific genes and downregulates the transcription of BEC-specific genes [103]. Multiple *Prox1* KO and overexpression studies have verified the critical role of PROX1 in lymphatic vessel development and maintenance [84,100,104]. Thus, the initiation of active *Prox1* transcription in ECs indicates LEC lineage specification. This observation led to an in-depth investigation of the transcriptional regulation of *Prox1*. 

### 3.2. Regulatory Networks of Epigenetic and Transcription Factors in Lymphatic Vessel Formation and Function

Transcriptome analysis and cellular imaging (for molecular colocalization) studies have revealed a high degree of correlation between PROX1 and other transcription factors involved in the development and maturation of lymphatic vasculature. Various transcription factors, including NR2F2, SOX18, GATA2, MAFB, FOXC2, NFATC1, and HHEX, are reported to be involved in the transcriptional regulation of PROX1. In turn, PROX1 is reported to be involved in the transcriptional regulation of LEC factors [104,105,106,107,108,109]. Additionally, several studies have reported the role of epigenetic modulators of transcription factors involved in lymph system development (Figure 2) [110,111,112,113,114]. The epigenetic factors that regulate chromatin conformational changes and histone modifications are indispensable to the optimal development of the lymphatic system. Analysis of gene expression and epigenetic modifications will provide valuable insights into the mechanisms underlying lymphatic development.

The binding motif of the transcription factor NR2F2 is located approximately 9.5 kb upstream of the open reading frame of mouse *Prox1* [106]. This sequence is conserved among various mammals [106]. β-Galactosidase staining and immunofluorescence analyses have demonstrated the colocalization of NR2F2 with PROX1 and LYVE1 in the LECs [105,106]. *Nr2f2* KO decreased the migration of PROX1-positive cells in the tissue around the cardinal vein of E11.5 and E13.5 mice. Additionally, the lymph sac was absent in *Nr2f2* KO mice [106]. However, *Nr2f2* deletion in PROX1-positive cells at E13.5 or later developmental stages did not result in major lymphatic defects. This indicates that NR2F2 promoted Prox1 expression only during early developmental stages [106]. BRG1, which regulates the expression of *NR2F2* [110], is a catalytic ATPase that constitutes the SWITCH/sucrose nonfermentable SWI/SNF-like complex. The SWI/SNF complex is an ATP-dependent chromatin remodeling complex that inhibits DNA-histone interactions. In the absence of BRG1, the *Nr2f2* promoter exhibits a highly compact structure, which results in the downregulation of *Nr2f2* expression. *Brg1* KO in endothelial cells recapitulated the genetic depletion of *Nr2f2* with the downregulation of lymphatic markers and the upregulation of arterial markers [110].

SOX18 binds to *Prox1* at two sites (1135–1130 bp and 813–808 bp) upstream of the transcription start site of *Prox1* [104]. The results of the luciferase reporter-gene assay with the mlEnd cell line and in vivo immunofluorescence experiments revealed that both binding sites must be intact for SOX18 to drive the expression of the reporter gene during LEC development [104]. *Prox1* expression was downregulated in the absence of functional SOX18. Conversely, the exogenously expressed *Sox18* restored the expression of *Prox1* [104]. *Sox18*-null mice completely lack LECs but contain other PROX1-positive cells, such as myocardial cells [104]. These results indicate that the transcription factor SOX18 specifically mediates the differentiation of BEC to LEC in the cardinal vein. DOT1L, a histone H3K79 methyltransferase, is involved in the transcriptional regulation of *Sox18* [111]. RNA-sequencing (RNA-seq) analysis revealed that the expression of *Sox18* is downregulated in LECs from E15.5 *Dot1l* KO mouse skin. Meanwhile, ChIP-seq revealed that H3K79 methylation was reduced in the gene body of *Sox18* in these LECs [111]. Similar ChIP-seq results were obtained for other LEC development-related genes, such as *Flt4*, *Ramp2*, and *Foxc2* [111].

MAFB promotes the transition of vascular ECs to LECs by upregulating the transcription of genes that promote LEC fate [107]. The binding motif of MAFB is in the first intron of *Prox1*. The induction of MAFB in primary LEC cell culture through VEGFR3, an upstream signal-transducing factor of MAFB, upregulated the expression of Prox1. MAFB is reported to bind to genes encoding other major factors regulating LEC specification, such as *Klf4*, *Nr2f2*, and *Sox18*. Short-interfering RNA-mediated *Mafb* knockdown downregulated the expression of *Prox1*. The role of MAFB in lymphatic development has been demonstrated using the back skin of *Mafb* KO E14.5 mouse embryos [107]. In these embryos, lymphatic vessel sprouting was delayed or incomplete.

Oscillatory shear stress (OSS) generated by lymph flow upregulates the expression levels of *GATA2*, *FOXC2*, and *NFATC1*, which regulate *PROX1* transcription in LECs [109]. Physical stimuli promote the transcription of *GATA2* [115]. GATA2, FOXC2, and NFATC1 may regulate PROX1 expression through their binding consensus sequences located close to each other at the first intron of *PROX1* (approximately 11 kb upstream from the transcription start site) [109]. FOXC2 and NFATC1 can form a complex. The downregulation of FOXC2 or NFATC1 results in impaired lymphatic vessel development. However, GATA2 interacts directly with neither FOXC2 nor NFATC1. This indicates that GATA2 transforms the enhancer region of *PROX1* into an ‘active’ state rather than directly inducing the transcription of *PROX1*. GATA2 regulates the opening of the *PROX1* promoter region, whereas HDAC3 regulates the promoter region of *Gata2* by inducing acetylation of H3K27 [112]. In the proposed model, HDAC3 is recruited to the intragenic enhancer of *Gata2* in response to OSS. Next, TAL1, GATA2, ETS1/2, and HDAC3 form a complex and promote the recruitment of EP300. Finally, the EP300-mediated accumulation of H3K27ac promotes the expression of GATA2.

The binding site of HHEX is located 800 bp upstream of the transcription start site of human *PROX1* [116]. The in vitro binding of HHEX to the *PROX1* promoter region was confirmed using cultured human umbilical vein endothelial cells. This suggests that HHEX directly regulates the expression of PROX1. *Hhex* knockdown downregulated the expression of *Prox1* in mouse and human LECs. *Hhex* KO in TIE2-positive cells impaired the proliferation, maturation, and sprouting of lymphatic vessels. HHEX is involved in the transcription of *Vegfc* and *Flt4*, which are involved in the development of lymphatic vessels. However, the binding of HHEX to the *Vegfc* and *Flt4* promoters could not be confirmed. This suggests that HHEX may indirectly regulate the transcription of *VEGFC* and *FTL4* and that the role of HHEX in lymphatic vessel development is mediated only by the activity of PROX1. Thus, the role of HHEX in LEC specification may be limited to the transcriptional regulation of PROX1.

The binding of PROX1 to target genomic DNAs induces the recruitment and transcriptional regulation of other co-factors [117,118]. Homeobox proteins share the helix-turn-helix (HTH) structure [119]. Similar to other homeobox proteins, the HTH structure of PROX1 enables it to directly bind to the major groove of DNA. The binding of PROX1 to DNA is followed by the recruitment of other epigenetic modifiers and transcription factors, such as the histone acetyltransferase EP300 [118]. Histone acetylation mediated by PROX1 and EP300 in the promoter region induces the unpacking of chromatin, the recruitment of other transcriptional factors, such as polymerase II, and the upregulation of the transcription of these genes. ChIP-seq analysis revealed that PROX1 binds to the promoter region of genes encoding proteins involved in lymphatic vessel development, such as VEGFR3, NRP2, SOX18, and PROX1 [106,115,120]. PROX1 targets *CPT1A*, which positively regulates the rate of fatty acid β-oxidation (FAO) [114]. Subsequently, the upregulated FAO induces acetyl coenzyme A production, which is utilized by EP300 for histone acetylation. The EP300/PROX1 complex promotes the histone acetylation of target gene promoters, including the promoter of *PROX1*. Thus, CPT1A is a part of the positive feedback loop of PROX1. PROX1 epigenetically regulates the expression of CYP7A1, which is involved in the bile acid synthesis pathway in the liver [121]. Co-immunoprecipitation (CoIP) studies with human hepatoblastoma cells (HepG2 cells) and GST-pull down assays performed using human embryonic kidney cells (HEK293T cells) revealed that the binding of PROX1 to the promoter represses the expression of target genes by recruiting the LSD1/NuRD complex, directly binding to the LSD1 unit, and promoting H3K4 hypomethylation. The role of CHD4, a subunit of the NuRD complex, in lymphatic development has been previously reported [113]. In the absence of VEGFC signaling, the Yap/Taz complex recruits the CHD4/NuRD complex to the *PROX1* promoter and regulates PROX1 expression in a negative feedback loop through the deactivation of the Hippo signaling pathway [122]. This regulatory mechanism is critical for the patterning of the lymphatic plexus.

Previous studies on the transcriptional regulation of PROX1 have provided useful insights into the key factors involved in LEC specification. However, the elucidation of the comprehensive transcription regulation mechanism of transcription factors and other LEC-specifying genes throughout embryonic development is currently ongoing. In the myriad questions to be answered, findings suggest that the promoters and enhancers of LEC specification-related genes play important roles in LEC specification. The TFs with major roles in LEC fate determination and proliferation interact with the promoter of Prox1 gene, and the modification of promoter accessibility by epigenetic regulation is believed to be a key factor in lymphatic system development. Future studies could focus on the dynamics of epigenetic alterations, which will further aid in understanding the lymphatic system development process.

## 4. Heterogeneity in LEC Origin

In the theories based on Sabin’s research, the cardinal vein is the sole source of the entire lymphatic system, and this was a widely accepted dogma [98]. However, several studies have demonstrated that cells of non-venous origin are involved in the formation of the lymphatic system. Studies demonstrated that non-CV EC-origin LEC progenitors also contribute to the formation of the lymph sac [120,123,124]. The additional sources of LEC progenitor include the intersomitic vessels and the superficial venous plexus [120,123]. In addition, an elegant 3D imaging study clearly showed a stepwise process of lymph sac formation: (1) LECs emerge from the CV, dorso-laterally migrate, and form a meshwork along with LECs originating from other vessels. (2) The LECs further undergo coalescence to form a lumen structure (called a peripheral longitudinal lymphatic vessel) at the first lateral intersegmental vessel branch. (3) LECs located close to the CV aggregate simultaneously and form a primordial thoracic duct (pTD) [123]. These cells emerge at different developmental stages, acquire tissue-specific functionalities, and complement the vein-derived lymphatic system [83,101,125,126,127,128]. Further studies on cells of different sources can aid in the elucidation of the molecular mechanisms underlying lymphatic development. The understanding of the transition from a non-venous source to LEC will enable the identification of factors required for LEC fate determination. The analysis of the heterogeneity of LEC from different tissues can potentially explain the regulation of the LEC developmental process. Some recent studies have successfully characterized LECs of non-venous origin (Figure 3).

The mesentery is a rich source of vasculature. The anatomy of the mesentery enables the analysis of vasculature functions without the need for sectioning. Additionally, a new concept of the non-venous developmental process has been defined using these vasculatures. Stanczuk et al. established a mouse lineage with heterozygous *Flt4*-null and heterozygous kinase-dead *Pik3ca* alleles (*Vegfr3*^lz/+^;*p110a*^D933A/+^), which exhibited poorly developed lymphatic vessels in the mesenteric root [101]. However, lymphatic vessels in the diaphragm and skin were not affected. Based on this evidence, the authors hypothesized a tissue-specific activity of the VEGFR3/PI3K axis. Further investigations revealed that the mesenteric lymph sac originates from a venous source. However, PROX1-positive and NRP2-positive ECs in the mesenteric membrane were discontinued from the mesenteric root. Clusters of these cells were observed between E13 and E13.5. These clusters formed the mesenteric lymphatic vasculature by E14.5. Lineage-tracing analysis confirmed that the source of the lymphatic vasculature was c-KIT-positive/VAV1-negative hemogenic endothelium. In the proposed model, the lymph vessels in the mesenteric root differentiate from the lymph sac, while the collecting lymph vessels in the mesenteric membrane and lymphatic capillaries in the intestine differentiate from the c-KIT lineage. The lack of DOT1L impairs the formation of the lymphatic vessel only when c-KIT-positive cells are affected [111]. This demonstrates the presence of tissue-origin specificity in the epigenetic mechanism that drives LEC specification. Similar to blood circulatory system development, Stanczuk et al. distinguished the formation of lymphatic vessels based on the sprouting from the lymph sac and the development of non-venous mesenteric lymphatic vessels and coined the term ‘lymphvasculogenesis.’ This study not only introduced the question of tissue-specific LEC heterogeneity but also identified the mammalian LEC population derived from a non-venous source, which has not been completely elucidated.

Martinez-Corral et al. discovered a novel source of LECs in the skin of mouse embryos [125]. The authors reported the discontinuation of the lymphatic vessels in the lumbar and cervical regions in the skin of E13.5 and E15.5 mice. During lineage-tracing of lumbar LEC, the authors used *Tie2-Cre*;*R26-mTmG* mice, which allowed the distinct identification of *Tie2*-expressing cells. In this mouse line, the activation of Cre recombinase permanently removed the sequence encoding tomato and labeled the cell with green fluorescent protein (GFP). Whole-mount analysis of the skin revealed LECs without GFP expression. Flow cytometry analysis of LYVE1-positive and PDPN-positive cells revealed that some LECs originated from non-*Tie2*-expressing cells. To identify the origin of LECs that do not originate from the lymph sac, the authors used *Prox1-CreER*;*R26-mTmG* mouse lineage injected with 4-Hydroxytamoxifen (4-OHT) at E12.5. The authors hypothesized that all LECs express GFP if all LECs were derived from the lymph sac and, otherwise, cells expressing tomato would be present in the lymphatic vasculature. In E17.5 mice, the presence of cells expressing tomato was observed at the sprouting points of the vasculature. Pichol-Thievend et al. proposed that these cells originate from the blood capillary plexus [126]. *Prox1* expression in the blood capillaries and some PROX1-positive cells at E13.5 has been observed in the lumen of the blood vasculature. The consensus finding between these two studies is that there is a source of LECs that is not derived from the lymph sac. However, the expression of *Tie2* in the originating cells was a disputed finding between the two studies. Pichol-Thievend et al. suggested that the discrepancy in the study by Martinez-Correl could be attributed to variability in *Tie2*-Cre-mediated recombination.

Klotz et al. reported that the origin of significant portions of cardiac lymphatic vessels was not from the lymph sac [83]. During mouse embryonic stages, the lymphatic vasculature on the heart was observed to emerge from the ventral side and postnatally complete a network with the cardiac vein and artery on the surface of the heart. In the hearts of E14.5 mice, the majority of the lymphatic tissue was labeled using *Tie2*-Cre-mediated labeling. The results of this experiment were consistent with the previous knowledge of the origin of LEC. However, approximately 19% of the lymphatic vessels in the heart did not exhibit *Tie2* expression and were of non-venous origin. These non-venous LECs did not originate from the epicardium, cardiac mesoderm, or cardiac neural crest. The authors hypothesized that these LECs originated from the TIE2-negative population of primitive hematopoietic cells within the yolk sac. This hypothesis was validated by verifying the expression of VAV1, PDGFRB, and CSF1R in these cells. The contribution of VAV1-positive cells to the cardiac lymphatic vessels was verified by the presence of lymphatic vessels after the ablation of *Prox1* expression in TIE2-positive cells and *Vav1*-Cre-mediated *Prox1*-EGFP expression in cardiac lymphatic vessels. Newer studies have proposed ISL1 as a marker for LEC precursor cells of the ventral cardiac lymphatic vessels. ISL1 is another factor that marks the LEC population of non-venous origin [127,128]. Additionally, ISL1-positive cells, which belong to a group of multipotent cell populations in the second heart field, are involved in the formation of the outflow tract and facial skin [127,131]. Interestingly, ISL1-positive cells originated exclusively from the ventral surface of the heart below the atrium. Further analysis revealed that the absence of LEC of the ISL1-positive lineage does not affect the dorsal lymphatic vessels. This was speculated to be due to the maturation of ISL1-positive LECs in response to a high concentration of retinoic acid in the ventricle region [128].

In contrast to the century-old perspective of examining LEC origin in the lymph sac or other tissue-specific sources during organ development, Oliver et al. proposed that the paraxial mesoderm is the origin of all LECs [130]. The authors reported that myogenic precursors in a somatic paraxial mesodermal cell line expressing PAX3 form the jugular lymph sac and its derivative lymphatic vessels, such as the lymphatic vessels in the heart, lung, and skin. Lineage-tracing using *Myf5-Cre* or *Mef2c-AHF-Cre* cell lines in combination with ROSA26^tdTomato^ can label PAX3-independent myogenic precursors from the paraxial mesoderm. These precursors served as LEC progenitors. The *Myf5*-*Cre* line contributes to the formation of lymphatic vessels in the lower jaw, meninges, ear skin, and a small portion of the lymph sac. The *Mef2c*-*AHF*-*Cre* line contributes to the formation of the second heart field. According to the ISL1 LEC study, these cells differentiate into LECs in the anterior jugular lymph sac, ventral cardiac LEC, and cervicothoracic dermis. The authors have claimed that they have captured the earliest steps in LEC differentiation before the involvement of the SOX18-NR2F2-PROX1 axis. This study warrants further lymphatics-related research focusing on the paraxial mesodermal lineage. Studies on the signaling network between the paraxial mesoderm lineage cells and the surrounding environment could provide useful insights into the molecular signals required for LEC specification. Additionally, the contribution of other PAX3-positive non-paraxial mesoderm sources to the LEC population could be investigated. The early lineage of mesenteric and intestinal lymphatic vessels has not been identified. Additional investigation of c-KIT-positive cells could reveal specific requirements for LEC specification.

The blood vessels and their tissue-specific functions have been extensively studied, and their tissue-specific functionalities have been highlighted. The identification of tissue-specific BECs led to the speculation of tissue-specific LECs. Thus, various studies have examined the different origins of LECs and endorsed their tissue specificity. Further studies are needed to examine the process of progenitors in the specification of definitive LEC fate, which will provide useful information on the mechanism of LEC induction. The increased number of HTS analyses and the discoveries of key factors have laid the foundation for the elucidation of LEC differentiation and the development of novel therapeutic strategies for LEC-related diseases, and the prospects are promising.

## 5. Conclusions

Lineage-tracing experiments and sequencing technologies with increased precision and functionality have enabled the elucidation of lymphatic system developmental processes. In the last decade, lineage-tracing studies have identified various key factors involved in lymphatic development and the contribution of previously unknown cells to the formation of LECs. Oliver et al. used lineage-tracing to demonstrate that priming for LEC specification occurs during mesoderm formation, before the emergence of the lymph sac [130]. However, some studies have raised concerns about the accountability of Cre-mediated recombination in lymphatic vessel-related studies [125,126]. Hence, additional validation steps must be considered in lineage-tracing experiments.

While lineage-tracing experiments focus on the origin of the LECs, sequencing experiments reveal the mechanism of LEC specification. HTS can identify the transcription profile of developing cells and indicate the importance of epigenetic regulation in developmental processes. Moreover, single-cell sequencing has increased in its precision and capacity to scope more cells [132] and databases, such as the EC atlas and MOCA, accumulating a vast amount of single-cell sequencing data [133,134]. However, these databases are not constructed specifically for LEC studies. The amount of LEC data included in these databases is insufficient for independent LEC analysis. Single-cell sequencing analysis of LECs from two or more different embryonic organs could provide useful information for the characterization of tissue-specific LEC development. In addition to the techniques currently used for developmental studies, recent studies have developed novel, cutting-edge techniques, such as artificial intelligence-based predictions of protein structure and function [135]. These techniques may aid in elucidating molecular interactions during LEC development and in establishing a complete LEC development map. Consequently, a complete LEC development map can contribute to the identification of effective therapeutic strategies for LEC-related diseases and determine the physiological advantage of enhanced lymphatic vessel functionalities.

## Figures and Tables

**Figure 1 cells-11-01692-f001:**
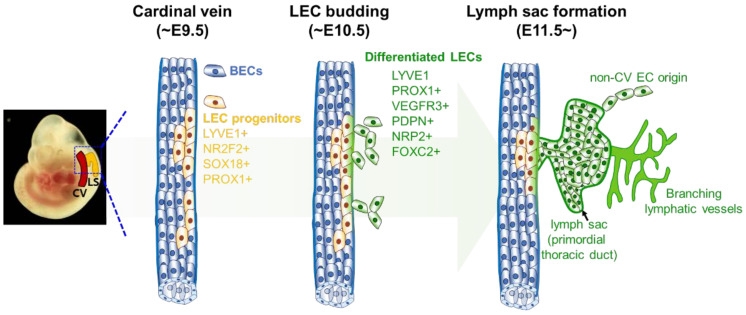
A schematic illustration of mouse lymphatic system development. During embryonic development (at approximately E9.5), a subset of blood endothelial cells in the cardinal vein expresses some initial lymphatic markers, such as LYVE1, NR2F2, SOX18, AND PROX1. The lymphatic endothelial cell (LEC) progenitors migrate into the lateral mesenchymal space, which is mediated by VEGFC signaling, and form primitive lymph sacs. The sprouting of LECs and the branching of lymphatic vessels from lymph sacs lead to the development of peripheral lymphatic vessels. CV: cardinal vein; LS: lymph sac; BEC: blood endothelial cell; LEC: lymphatic endothelial cell.

**Figure 2 cells-11-01692-f002:**
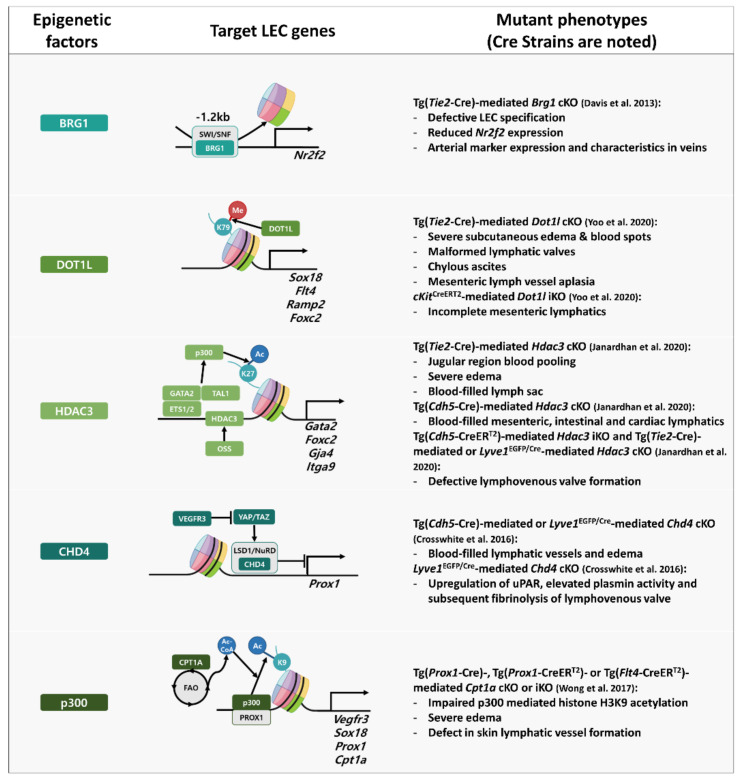
Epigenetic factors that regulate the transcription of LEC-associated factors that modulate chromatin conformation or the recruitment of cofactors [110,111,112,113,114]. Arrowheads represent the initiation of transcription or promotion of acetylation or methylation and flat-headed lines represent the repression of protein function or transcription. LEC: lymphatic endothelial cell; kb: kilobasepair; KO: knock-out; cKO: conditional knock-out; iKO: inducible knock-out; K79-Me: methylation on 79th lysine (K) residue of histone H3; K27-Ac: acetylation on 27th lysine (K) residue of histone H3; OSS: oscillatory shear stress; K9-Ac: acetylation on 9th lysine (K) residue of histone H3; FAO: fatty acid oxidation (in mitochondria); Ac-CoA: acetyl coenzyme A.

**Figure 3 cells-11-01692-f003:**
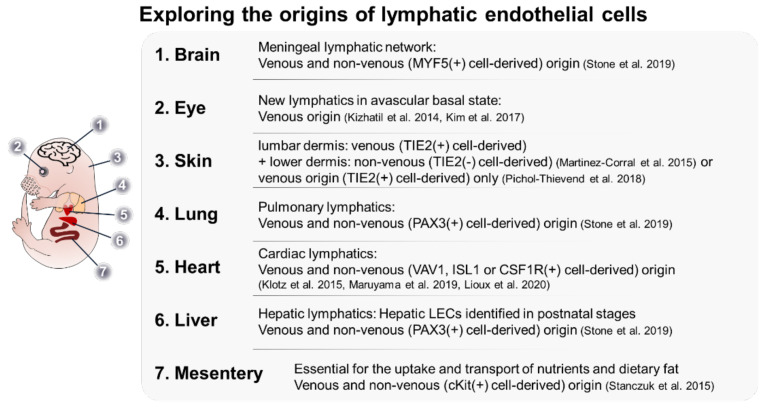
Current models of the origins of organ-specific LECs in mice. Several studies have used various lineage-tracing methods to demonstrate that the diverse non-venous source-derived lymphatic progenitors contribute to the development of tissue-specific lymphatic vessels [83,95,101,125,126,127,128,129,130].

**Table 1 cells-11-01692-t001:** Genetic disorders associated with primary lymphedema.

Genes	Disorders	Phenotype	OMIM	Reference
*VEGFR3*	Nonne–Milroy disease	- Congenital bilateral lower limb lymphedema - Chylous ascites - Apparent at birth (Type I)	153,100	(Butler et al., 2007, Butler et al., 2009) [31,32]
*VEGFC*	Congenital primary lymphedema of Gordon	- Similar to VEGFR3 phenotype	615,907	(Balboa-Beltran et al., 2014, Gordon et al., 2013) [33,34]
*GJC2*	Late-onset autosomal dominant lymphedema	- At birth or early childhood - Impact on all extremities	613,480	(Ferrell et al., 2010) [35]
*FOXC2*	Lymphedema–distichiasis syndrome	- Distichiasis - Leg lymphedema - Physiological number of lymphatic vessels but dysfunctional lymphatic drainage	153,400	(De Niear et al., 2018, Rezaie et al., 2008) [20,21]
*SOX18*	Hypotrichosis-lymphedema-telangiectasia-renal defect syndrome and hypotrichosis-lymphedema-telangiectasia syndrome	- Rare - Absence of eyebrows and eyelashes - Hypotrichosis, lymphedema, telangiectasia, and renal features	137,940 607,823	(Irrthum et al., 2003, Moalem et al., 2015) [22,23]
*EPHB4*	Autosomal dominant lymphatic-related hydrops fetalis (LRHF)	- Non-immune LRHF in utero, resulting in embryonic lethality	617,300	(Martin-Almedina et al., 2016) [36]
*CCBE1*	Hennekam-lymphangiectasia-lymphedema syndrome Type 1	- Severe defects, including intestinal lymphangiectasias, mental retardation, and facial dysmorphism	235,510	(Connell et al., 2010) [37]
*FAT4*	Type 2		616,006	(Alders et al., 2014) [38]
*ADAMTS3*	Type 3		618,154	(Brouillard et al., 2017) [39]
*FBXL7*	Hennekam-lymphangiectasia-lymphedema syndrome		-	(Boone et al., 2020) [40]
*GATA2*	Emberger syndrome	- Myeloblastic leukemia	614,038	(Emberger et al., 1979, Mansour et al., 2010) [24,25]
*CELRS1*	Late-onset hereditary lymphedema	- Non-syndromic - Limited to females	-	(Gonzalez-Garay et al., 2016) [41]
*KIF11*	Microcephaly-chorioretinopathy-lymphedema syndrome	- Microcephaly, chorioretinopathy, lymphedema, or mental retardation	152,950	(Birtel et al., 2017) [42]
*PIEZO1*	Generalized lymphatic dysplasia	- Uniform widespread edema - Intestinal and/or pulmonary lymphangiectasia - Pleural effusions, chylothorax, and/or pericardial effusions	616,843	(Fotiou et al., 2015) [43]
*RASA1*	Capillary malformation-arteriovenous malformation/lymphedema	- Capillary malformations and arteriovenous malformations	608,354	(Revencu et al., 2013) [44]
*PTPN14*	Choanal atresia-lymphedema	- High-arched palate, hypoplastic nipples, and mild pectus excavatum	613,611	(Hiramatsu et al., 2017, Qazi et al., 1982) [45,46]
*CALCRL*	Hydrops fetalis	- Lymphatic dysplasia - Non-immune	114,190	(Mackie et al., 2018) [47]
*ITGA9*	Fetal chylothorax	- Missense mutation causes lymphedema in fetuses	-	(Ma et al., 2008) [48]
*RELN*	Cerebellar hypoplasia	- Neonatal lymphedema - Chylous ascites	-	(Hong et al., 2000) [49]

Modified from Gordon et al., 2020, and Oliver et al., 2020 [3,19].

## Data Availability

Not applicable.

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
