# Peer review of "Role of Transcriptional and Epigenetic Regulation in Lymphatic Endothelial Cell Development"

_cells, 2022, doi:10.3390/cells11101692_

Round 1

Reviewer 1 Report

Title: Role of Transcriptional and Epigenetic Regulations in Lymphatic Endothelial Cell Development

Summary:

The manuscript “Role of Transcriptional and Epigenetic Regulations in Lymphatic Endothelial Cell Development” by Hyeowoon et al. gives an overview of previous knowledge about lymphatic development and more recent findings about epigenetic influences and non-lymph sac derived LECs. Additionally, it is giving a summary of pathological conditions with lymphatic involvement.

The reviewer would like to positively highlight Figures 1 and 3, which are highly comprehensive and clear. Furthermore, the historical excurse on Florence Sabin was not only interesting but gave the reader an easy entry into the topic.

Major comments:

The manuscript gives a variety of information and outlines scientific discourse in the field. However, the reviewer would advise the authors to rethink the structure of the presented information. The main concern are the clinical implications at the very end of the paper. The reviewer has found them to be well comprehensible without the information of the section prior and is therefore wondering if the paper would not benefit from rearranging this section to the beginning of the paper. Starting with a concrete clinical need usually draws the readers attention and in the very beginning shows the relevance of the topic.

Furthermore reviewer would advise to insert more sub headings, especially in the section “Key Transcription Factors of LECs and Epigenetic Regulation of their Transcription”. This part is highly informative, but it is hard for the reader to follow the thoughts as the structure is not always clear. Another way to improve this section would be a revision of Figure 2. From reviewers understanding the authors describe in the following paragraphs certain factors of the graphic. By dividing the Figure into a,b,c,… and adding a reference to the text this could make it easier to follow.

Further it is unclear to me how the Tie2 KO and the epigenetic factors relate in the Figure, especially as the same KO is present for different epigenetic factors. Overall, reviewer does not consider the graphic to be comprehensive as a standalone figure. There are abbreviations such as Me or Ac without further explanation. The abbreviations are clear for readers familiar with the field but I would kindly ask the authors to include the explanation for the abbreviations. 

The conclusion is focusing a lot on the advantages of single-cell seq, which has not been the main content of the review. Perhaps the main focus of the conclusion should be re-thought and/or single-cell seq in the field should be highlighted previously.

Minor Comments:

Authors nicely illustrate the initial steps of early lymphangiogenesis in the mouse and illustrate LEC progenitors leaving the cardinal vein giving rise to early lymphatic structures. The reviewer would emphasize to include a short statement/illustration that early LECs forming the initial lymph sac are not only cardinal vein-derived but also additional venous vessels. As various studies have investigated the formation of the initial lymphatic vasculature and showed that the 3D-structure of initial lymph vessels are rather tube-like than sac like (Dartsch et al. 2014 MCB, Yang, Oliver et al, 2012 Blood, Haegerling et al. 2013 EMBO J,), the reviewer would emphasize to include the term ‘primordial thoracic duct’ beside the term lymph sac. From the reviewer’s perspective, this is a relevant due to a more precise description of developmental processes leading to the final/definitive thoracic duct.

From line 65 there is a lot of detail about studies and methods in the Introduction about epigenetics. Perhaps this information could rather be placed in a later paragraph together with the Epigenetic aspect of the paper, as the review is not only engaging with epigenetics.

GATA2 is once referred to as a transcriptional factor (l.114), but then listed as an epigenetic factor in Figure 2. Perhaps a clarification of what an epigenetic factor is would be helpful, as the paper is dedicating large parts to epigenetics.

The reviewer would advise to add a short summary sentence at the beginning of the Conclusion. The reader is with his/her thoughts currently with diseases and is now all of a sudden jumping back to molecular details (or change that order).

The authors list various publication on KO mice and lymphatic phenotypes, however, a few citations are insufficient. Additional citations and referenced manuscripts would be helpful for the reader to get a broader overview on lymphatic phenotypes. E.g. important contributions on VE-Cadherin and Lymph vessel specific KO (Carmeliet 1999 Cell, Haegerling et al. 2018 EMBO J) as well as on CCBE1 KO (Bos et al. 2009) should be added.

Author Response

The authors thank the reviewers very much for their valuable comments and support. The following is our point-by-point responses to their comments.

Reviewer 1

The manuscript “Role of Transcriptional and Epigenetic Regulations in Lymphatic Endothelial Cell Development” by Hyeowoon et al. gives an overview of previous knowledge about lymphatic development and more recent findings about epigenetic influences and non-lymph sac derived LECs. Additionally, it is giving a summary of pathological conditions with lymphatic involvement.

The reviewer would like to positively highlight Figures 1 and 3, which are highly comprehensive and clear. Furthermore, the historical excurse on Florence Sabin was not only interesting but gave the reader an easy entry into the topic.

Major comments:

The manuscript gives a variety of information and outlines scientific discourse in the field. However, the reviewer would advise the authors to rethink the structure of the presented information. The main concern are the clinical implications at the very end of the paper. The reviewer has found them to be well comprehensible without the information of the section prior and is therefore wondering if the paper would not benefit from rearranging this section to the beginning of the paper. Starting with a concrete clinical need usually draws the readers attention and in the very beginning shows the relevance of the topic.

Response: Thank you so much for the constructive comment. As the reviewer suggested, we have moved the section to the beginning of the manuscript.

Furthermore reviewer would advise to insert more sub headings, especially in the section “Key Transcription Factors of LECs and Epigenetic Regulation of their Transcription”. This part is highly informative, but it is hard for the reader to follow the thoughts as the structure is not always clear. Another way to improve this section would be a revision of Figure 2. From reviewers understanding the authors describe in the following paragraphs certain factors of the graphic. By dividing the Figure into a,b,c,… and adding a reference to the text this could make it easier to follow.

Response: As the reviewer suggested, we have added subheadings into the section. The newly added subheading are “3.1 Historical aspects of lymphatic vessel development” and “3.2 Regulatory networks of epigenetic and transcription factors in lymphatic vessel formation and function”

Further it is unclear to me how the Tie2 KO and the epigenetic factors relate in the Figure, especially as the same KO is present for different epigenetic factors. Overall, reviewer does not consider the graphic to be comprehensive as a standalone figure. There are abbreviations such as Me or Ac without further explanation. The abbreviations are clear for readers familiar with the field but I would kindly ask the authors to include the explanation for the abbreviations. 

Response: We apologize for any confusion we caused. We meant a Tie2-Cre transgenic line used in the conditional KO studies. Therefore, Tie2 KO means Tg(Tie2-Cre)-mediated Brg1, Dot1L… conditional KOs. We have changed the Figure 2 accordingly. We have also added full descriptions (K79-Me: methylation on 79th lysine (K) residue of histone H3, K27-Ac: Acetylation on 27th lysine residue of histone H3, and so on) of the abbreviations in the figure legend.

The conclusion is focusing a lot on the advantages of single-cell seq, which has not been the main content of the review. Perhaps the main focus of the conclusion should be re-thought and/or single-cell seq in the field should be highlighted previously.

Response: We have rearranged the content about single cell analysis.

Minor Comments:

Authors nicely illustrate the initial steps of early lymphangiogenesis in the mouse and illustrate LEC progenitors leaving the cardinal vein giving rise to early lymphatic structures. The reviewer would emphasize to include a short statement/illustration that early LECs forming the initial lymph sac are not only cardinal vein-derived but also additional venous vessels. As various studies have investigated the formation of the initial lymphatic vasculature and showed that the 3D-structure of initial lymph vessels are rather tube-like than sac like (Dartsch et al. 2014 MCB, Yang, Oliver et al, 2012 Blood, Haegerling et al. 2013 EMBO J,), the reviewer would emphasize to include the term ‘primordial thoracic duct’ beside the term lymph sac. From the reviewer’s perspective, this is a relevant due to a more precise description of developmental processes leading to the final/definitive thoracic duct.

Response: As the reviewer suggested, we have added the following. “Studies demonstrated that non-CV EC-origin LEC progenitors also contribute to the formation of the lymph sac. The additional sources of LEC progenitor include the intersomitic vessels and the superficial venous plexus. In addition, an elegant 3D imaging study clearly showed a stepwise process of lymph sac formation: (1) LECs emerged from CV dorso-laterally migrate and form a meshwork along with LECs originating from other vessels. (2) The LECs further undergo coalescence to form a lumen structure (called a peripheral longitudinal lymphatic vessel) at the first lateral intersegmental vessel branch. (3) LECs located close to the CV aggregate simultaneously and form a primordial thoracic duct (pTD).”

From line 65 there is a lot of detail about studies and methods in the Introduction about epigenetics. Perhaps this information could rather be placed in a later paragraph together with the Epigenetic aspect of the paper, as the review is not only engaging with epigenetics.

Response: We have added the following sentence for the paragraph and rearranged the content following the quoted sentence to provide a more general idea of the review. “For example, genetic studies also revealed that that molecules such as VE-Cadherin, a cell-cell adhesion molecule, and CCBE1, a factor involved in activation of VEGFC, are essential for lymphangiogenesis.”

GATA2 is once referred to as a transcriptional factor (l.114), but then listed as an epigenetic factor in Figure 2. Perhaps a clarification of what an epigenetic factor is would be helpful, as the paper is dedicating large parts to epigenetics.

Response: We have removed GATA2 in the Figure 2 as it is a TF.

The reviewer would advise to add a short summary sentence at the beginning of the Conclusion. The reader is with his/her thoughts currently with diseases and is now all of a sudden jumping back to molecular details (or change that order).

Response: We have relocated the diseases section to the beginning of the manuscript.

The authors list various publication on KO mice and lymphatic phenotypes, however, a few citations are insufficient. Additional citations and referenced manuscripts would be helpful for the reader to get a broader overview on lymphatic phenotypes. E.g. important contributions on VE-Cadherin and Lymph vessel specific KO (Carmeliet 1999 Cell, Haegerling et al. 2018 EMBO J) as well as on CCBE1 KO (Bos et al. 2009) should be added.

Response: As wrote above, at line62, we have added the following sentence to highlight the studies. “Genetic studies also revealed that that molecules such as VE-Cadherin, a cell-cell adhesion molecule, and CCBE1, a factor involved in activation of VEGFC, are essential for lymphangiogenesis.”

Reviewer 2 Report

The authors described "Role of Transcriptional and Epigenetic Regulations in Lymphatic Endothelial Cell Development" in Review style. As they pointed out,  research on embryonic development is critical for devising therapeutic strategies for lymphatic diseases. In addition, the elucidation of the molecular mechanisms underlying the formation of the lymphatic system in the early stages will enable the development of useful strategies for the reconstitution of the optimal functioning of the lymphatic system. I have one suggestion to improve this manuscript.

Recent studies investigated "intussusceptive lymphangiogenesis" like intussusceptive angiogenesis in lymphedema regions. Díaz-Flores et al. suggested a possible molecular mechanism of intussusceptive lymphangiogenesis in the developing lymph node by which a high abundance of VEGF-C whole lymph node cells, without VEGF-C gradient, results in the nonsprouting engulfment of the lymph node anlage by LECs (Ann. Anat. Anat. Anz. 2019). Also, Ogino R et al. reported addipose derived stem cells transplantation accelerated LEC proliferation, increased lymphatic vessel numbers, and mitigated fibrosis of the surrounding interstitial tissue via intussusceptive lymphangiogenesis (Int. J. Mol. Sci. 2020)

The authors should discuss intussusceptive lymphangiogenesis in section 4.

Author Response

The authors thank the reviewers very much for their valuable comments and support. The following is our point-by-point responses to their comments.

Reviewer 2:

The authors described "Role of Transcriptional and Epigenetic Regulations in Lymphatic Endothelial Cell Development" in Review style. As they pointed out, research on embryonic development is critical for devising therapeutic strategies for lymphatic diseases. In addition, the elucidation of the molecular mechanisms underlying the formation of the lymphatic system in the early stages will enable the development of useful strategies for the reconstitution of the optimal functioning of the lymphatic system. I have one suggestion to improve this manuscript.

Recent studies investigated "intussusceptive lymphangiogenesis" like intussusceptive angiogenesis in lymphedema regions. Díaz-Flores et al. suggested a possible molecular mechanism of intussusceptive lymphangiogenesis in the developing lymph node by which a high abundance of VEGF-C whole lymph node cells, without VEGF-C gradient, results in the nonsprouting engulfment of the lymph node anlage by LECs (Ann. Anat. Anat. Anz. 2019). Also, Ogino R et al. reported addipose derived stem cells transplantation accelerated LEC proliferation, increased lymphatic vessel numbers, and mitigated fibrosis of the surrounding interstitial tissue via intussusceptive lymphangiogenesis (Int. J. Mol. Sci. 2020). The authors should discuss intussusceptive lymphangiogenesis in section 4.

Response: The authors thank the reviewer very much for the comment. As the reviewer suggested, we have added the following sentence in the revision. “Another potential molecular mechanism involved in the formation of lymphedema has been suggested by Díaz-Flores et al. Their recent studies have shown that during human LN development, intussusceptive lymphangiogenesis is induced by highly abundant and evenly distributed VEGFC. In turn, intussusceptive lymphangiogenesis has been found to participate in the formation of the meshwork of processes in LN sinuses. Their studies provided the foundation for explanation of the role of intussusceptive lymphangiogenesis in clinical cases of lymphedema. Furthermore, Ogino et al. showed that transplantation of adipose-derived stem cells accelerated LEC proliferation, increased lymphatic vessel numbers, and mitigated fibrosis of the surrounding interstitial tissue via intussusceptive lymphangiogenesis.”

Reviewer 3 Report

Hyeonwoo La et al. overviewed the roles of transcriptional and epigenetic regulations in lymphatic endothelial cell, which provided a comprehensive elaboration of key factors involved in lymphatic system development. However, there are several details need to be replenished.Major concerns:1.      In Figure 2, authors reviewed several epigenetic factors which regulated the transcription of LEC-associated factors. However, authors focused on Prox1 in Description. How about other epigenetic factors, including BRG1, DOT1L, HDAC, GATA2 or CHD4? Thus, authors should make some changes, which aims for harmony of figure and description.2.      In the part of “4. Diseases Associated with the Lymphatic System”, authors stated that inhibition of lymphangiogenesis could suppress tumor metastasis. However, recent studies indicated that VEGFC treatment with lymphatic expansion could enhance efficacy of immunotherapy or radiotherapy of glioma (Nature. 2020 Jan;577(7792):689-694; Cell Res. 2020 Mar;30(3):229-243; Cell Res. 2022 Mar 17. doi: 10.1038/s41422-022-00639-5.).3.    Most of the description of gene name is inconsistent, especially in part 2 Key Transcription Factors of LECs and Epigenetic Regulation of their Transcription. Such as, “Prox1-positive cells” on line 132 and “PROX1-positive cells” on line 135, “the expression of NR2F2” on line 137 and “the downregulation of Nr2f2 expression” on line 141, and so on.4.    In the introduction (line 376-386), the authors state that “Lymphatic vessels enable the transportation of activated antigen-presenting cells secondary lymph organs (lymph nodes) during adaptive immune responses.” This is right but not complete. In addition to transport APCs, LECs also participate in antigen presentation. The authors should discuss more in this area.5.    In the introduction (line 400-411), when the authors discuss the relations between lymphatic vessels and cancer, they only emphasize the effect of promoting tumor metastasis, which is one-sided. Some articles have demonstrated that lymphatic vessels could also promote anti-tumor immunity. It would be more appropriate, if the authors focus on the dual character of lymphatic vessels in cancer. Minor concerns:1.     Expanded form of abbreviations should be mentioned at first use in the body of the manuscript.2.     Please check Table 1, as there was an unknown gene with no reference.3.      In the introduction (line75-76): “and various lymphatic organs, such as the lymph nodes, bone marrow, and spleen”, the singular and plural forms should be consistent.4.      In the description of Fig.1 (line 109): “Vegfc” should be “VEGFC”.5.      In the introduction (line 115): the transitional word “Conversely” is inaccurately used.

Author Response

The authors thank the reviewers very much for their valuable comments and support. The following is our point-by-point responses to their comments.

Reviewer 3:

Major concerns:

In Figure 2, authors reviewed several epigenetic factors which regulated the transcription of LEC-associated factors. However, authors focused on Prox1 in Description. How about other epigenetic factors, including BRG1, DOT1L, HDAC, GATA2 or CHD4? Thus, authors should make some changes, which aims for harmony of figure and description.

Response: The epigenetic factors that the reviewer mentioned have already been included in the Figure 2 and section of “3. Key Transcription Factors of LECs and Epigenetic Regulation of their Transcription”. As you might be aware, we tried to emphasize regulatory networks between the epigenetic factors and TFs in the section.

In the part of “4. Diseases Associated with the Lymphatic System”, authors stated that inhibition of lymphangiogenesis could suppress tumor metastasis. However, recent studies indicated that VEGFC treatment with lymphatic expansion could enhance efficacy of immunotherapy or radiotherapy of glioma (Nature. 2020 Jan;577(7792):689-694; Cell Res. 2020 Mar;30(3):229-243; Cell Res. 2022 Mar 17. doi: 10.1038/s41422-022-00639-5.)

Response: Thank you for the comment. As you suggested, we have added the following sentence. “On the other hand, recent studies have shown that VEGFC treatment with lymphatic expansion could enhance anti-tumor immunity and efficacy of immunotherapy or radiotherapy of glioma, suggesting a dual function of lymphatic system on tumor metastasis in a context-dependent manner.” We have also included the references in the revision.

Most of the description of gene name is inconsistent, especially in part 2 Key Transcription Factors of LECs and Epigenetic Regulation of their Transcription. Such as, “Prox1-positive cells” on line 132 and “PROX1-positive cells” on line 135, “the expression of NR2F2” on line 137 and “the downregulation of Nr2f2 expression” on line 141, and so on.

Response: The inconsistency in the manuscript has been checked and edited. The links to the guidelines we followed can be found underneath. (http://www.informatics.jax.org/mgihome/nomen/index.shtml; http://genenames.org/about/guidelines/)

In the introduction (line 376-386), the authors state that “Lymphatic vessels enable the transportation of activated antigen-presenting cells secondary lymph organs (lymph nodes) during adaptive immune responses.” This is right but not complete. In addition to transport APCs, LECs also participate in antigen presentation. The authors should discuss more in this area.

Response: Thank you for the reviewer’s comment. We have added the following sentence in the revision. “LECs also participate in the process of inflammatory response regulation by mediating the antigen presentation and inducing the CD4 T cell tolerance. Recent studies suggest that LECs present peptide:MHC-II complexes acquired from dendritic cells (DCs) or participate in the process of antigen presentation of DCs by providing various peripheral tissue antigens (PTAs) to induce CD4 T cell tolerance.”

In the introduction (line 400-411), when the authors discuss the relations between lymphatic vessels and cancer, they only emphasize the effect of promoting tumor metastasis, which is one-sided. Some articles have demonstrated that lymphatic vessels could also promote anti-tumor immunity. It would be more appropriate, if the authors focus on the dual character of lymphatic vessels in cancer.

Response: Thank you for the comment. We have added the following sentence in the revision. “On the other hand, immune modulation by LV is also critical for trafficking of DC and initiating anti-tumor adaptive immunity (i.e. T cell responses), suggesting dual function of lymphatics in tumor metastasis depending on tumor types and tumor progression.”

Minor concerns:

  1. Expanded form of abbreviations should be mentioned at first use in the body of the manuscript.

Response: We have added full descriptions of the abbreviations in the revision.

  1. Please check Table 1, as there was an unknown gene with no reference.

Response: We have removed it as responsible gene(s) for it is unknown.

  1. In the introduction (line75-76): “and various lymphatic organs, such as the lymph nodes, bone marrow, and spleen”, the singular and plural forms should be consistent.

Response: We have changed “lymph nodes” to “LN”

  1. In the description of Fig.1 (line 109): “Vegfc” should be “VEGFC”.

Response: We have changed it.

  1. In the introduction (line 115): the transitional word “Conversely” is inaccurately used.

Response: We have changed it to “In turn”.

Round 2

Reviewer 3 Report

The authors have fully addressed my concerns and comments.